# A Comprehensive Evaluation of Metabolomics Data Preprocessing Methods for Deep Learning

**DOI:** 10.3390/metabo12030202

**Published:** 2022-02-24

**Authors:** Krzysztof Jan Abram, Douglas McCloskey

**Affiliations:** Novo Nordisk Foundation Center for Biosustainability, Technical University of Denmark, 2800 Lyngby, Denmark; krzyja@biosustain.dtu.dk

**Keywords:** metabolomics, deep learning, preprocessing

## Abstract

Machine learning has greatly advanced over the past decade, owing to advances in algorithmic innovations, hardware acceleration, and benchmark datasets to train on domains such as computer vision, natural-language processing, and more recently the life sciences. In particular, the subfield of machine learning known as deep learning has found applications in genomics, proteomics, and metabolomics. However, a thorough assessment of how the data preprocessing methods required for the analysis of life science data affect the performance of deep learning is lacking. This work contributes to filling that gap by assessing the impact of commonly used as well as newly developed methods employed in data preprocessing workflows for metabolomics that span from raw data to processed data. The results from these analyses are summarized into a set of best practices that can be used by researchers as a starting point for downstream classification and reconstruction tasks using deep learning.

## 1. Introduction

Although the genome defines the possible phenotypes for a cell or organisms, and the transcriptome and proteome describe what elements are active given a particular environmental condition, it is the metabolome, or the complement of all small molecules called metabolites, or that are the most sensitive to describing differences in physiology. In addition, a large portion of the work and resources that are dedicated to analytical labs in pharmaceutical, biotech, forensic, clinical, and diagnostic sectors across the world are focused on metabolomic assays for screening, quality control, or general phenotyping applications. Hence, metabolomic data provides a well-suited input modality for machine-learning applications because it is readily available, information rich, and can be obtained through established infrastructure in a plethora of life science domains.

Machine learning, and more specifically, deep learning, has greatly advanced over the past decade owing to advances in algorithmic innovations including back propagation [1], advanced optimizers [2], hardware [3], and domain specific techniques [4,5,6] to name a few. These advances have culminated in a slew of breakthroughs in computer vision, natural-language processing, and board/video games that in many cases best human performance [7]. More recently, deep learning has made advances in the life sciences in the domains of genomic variant calling and labeling [8], sequence-to-structure prediction [9,10], and gene, protein, and metabolite analysis [11,12,13]. The latter applications, focusing on the analysis and interpretation of gene, protein, and metabolite data, are particularly difficult for deep learning due to the lack of suitable training data, data heterogeneity, and high correlations among features. In particular, metabolomics datasets often suffer from low sample sizes and batch effects between datasets due to the complexity of sample collection and analytical protocols and methods [14,15], and large numbers of missing values due to variations in amounts of raw material analyzed that lead to some samples having levels of metabolites that are below the instrument limits of detection.

One of the first steps in any machine-learning workflow is the preprocessing of data. Data preprocessing is performed to cast the raw data into a machine-friendly space that is amenable to ingestion by the particular machine-learning algorithm. Examples of common data pre-process steps in metabolomics include normalization, standardization, and missing value imputation [16,17,18]. Each of these steps can have a profound impact on the final results [19,20,21,22]. Similar to other machine-learning algorithms, the data preprocessing steps can have a large impact on deep-learning algorithm performance. However, a comprehensive evaluation of metabolomics data preprocessing steps for deep learning has not been conducted.

In this study, a comprehensive evaluation of Omics data preprocessing steps is conducted to evaluate the effect on classification and reconstruction performance using prototypical datasets that would be considered unsuitable for deep learning based on sample size. The data preprocessing evaluated include biomass normalization, data calibration, missing value imputation, and various transformations and normalization/standardization techniques. Both quantitative and non-quantitative data with and without known batch effects and with balanced and unbalanced sample numbers are evaluated. Multiple evaluation metrics are also considered to provide a more complete evaluation of classification and reconstruction performance. During the evaluation, approximately five thousand models were trained requiring close to six months of continuous compute time. A set of best practices based on these results are provided in the Conclusion.

## 2. Results and Discussion

### 2.1. Overview of the Metabolomics Workflow, Datasets, and Experiments

Machine-learning performance using metabolomics, lipidomics, proteomics, and other analytical chemistry-based Omics data that have been acquired using targeted or non-targeted assays depend strongly upon the data-preprocessing steps used. Figure 1A depicts the steps and options in a prototypical machine-learning data-preprocessing workflow for analytical chemistry-based Omics data. Besides the influences of biological and operator variability when running the experiment and generating the samples, batch effects can be introduced during data acquisition, during any automated or manual QC/QA steps, or during data calibration (red lettering in Figure 1A). Following data calibration, one has a large design space to modify the data prior to machine-learning model training. The design space includes biomass normalization, missing value imputation (Figure 2), data normalization, and online normalization. Options for each of the steps that are explored in this study are provided in Figure 1A. Although the options tested are not exhaustive to what has been explored by others, these provide a well-rounded set of unique and well-accepted options used in the literature.

Batch effects and class imbalances are common attributes of analytical chemistry Omics datasets. For this reason, two different datasets with either of these attributes were used to better understand the influence of batch effects and class imbalances on classification and reconstruction accuracy when investigating the influence of each of the steps in the data-preprocessing pipeline. Dataset 1 is class balanced, but has batch effects resulting from the fact that the same samples were acquired using two different methods (Figure 1B). Dataset 2 has no discernible batch effects, but has several class imbalances owing to practical considerations when performing the original experiment (Figure 1C). These two datasets were used in the below sections.

The influence of data calibration, biomass normalization, missing value imputation, and data normalization on classification accuracy are explored in the below sections.

### 2.2. Data Calibration and Biomass Normalization Have Subtle Influence on Classification Accuracy and Training Speed

Calibrating and normalizing acquired signal intensity from the analytical instrument to metabolite abundances is a multi-step process that involves transforming raw ion counts to absolute or relative concentrations. The isotope dilution mass spectrometry (IDMS) was employed for the datasets used in this study and encompasses the main steps required for any calibration and normalization method used in analytical chemistry. In brief, the steps for data calibration used here included the following:raw ion counts for each metabolite are integrated into peaks and reported as the peak apex or peak area (Intensities),the intensities of the metabolite are normalized to the intensities of their corresponding internal standard (Ratios), andconcentrations are calculated for each metabolite based on an externally ran calibration curve consisting of standards with known amounts (Concs).

The concentrations determined in step 3 are then normalized to absolute or relative amounts by dividing by the sample biomass where the biomass can be expressed as e.g., dry weight for cell cultures.

The influence of biomass normalization in combination with data calibration steps 1–3 were tested. No significant differences in classification accuracy nor training speed was observed between models trained with or without biomass normalization for datasets 1 and 2 (Figure 3C,D and Appendix A). However, depending upon the transformation option used, significant differences in classification accuracy, but not training speed, were observed at different steps in the calibration process (Figure 3C,D and Appendix A). In particular, Intensities showed poor performance and generalization to the test data set compared to Ratios or Concs (Figure 3C,D and Appendix A). These findings indicate that the model is able to account for differences in sample biomass, but performance can degrade when sample handling and data acquisition effects are not normalized.

### 2.3. Sampling-Based Methods Provide a Simple and Powerful Missing Value Imputation Strategy

Missing values are common when using high throughput analytical methods used in generating-Omics data. In metabolomics, missing values are most often a result of the metabolite being below the level of detection in some replicates but not others or if there happens to be a high experimental or analytical variance in the measurement of the metabolite resulting in an outlier that is removed during the raw data QC/QA process. Many statistical methods do not allow for missing values, and therefore researchers turn to various strategies to fill in missing values in their datasets. The following imputation strategies were explored and are visualized in Figure 2:Filling with zeros (FillZero),Imputing missing values using a probabilistic model (ImputedAmelia, see methods),Sampling, andCasting the data to mass action ratios (MARs) and sampling the data to compute the MARs (Figure 2D,E).

The influence of missing value imputation methods were tested. Significant differences in classification accuracy and training speed between the different missing value imputation methods was observed for models trained using datasets 1 and 2 (Figure 3C,D and Appendix A). In general, Sampling showed the highest classification accuracy on both training and test data splits followed closely by MARs and then ImputedAmelia and FillZero (Figure 3C,D and Appendix A). In general, Sampling showed the fastest training convergence followed closely by MARs and then ImputedAmelia and FillZero (Figure 3C,D and Appendix A). It should be noted that depending upon the transformation option used, the ranking of Sampling and MARs can differ with respect to classification accuracy on either the training or test data splits. These results indicate that sampling based strategies (i.e., Sampling and MARs) appear to prevent overfitting, improve training convergence speed, and can alleviate the need for sample handling and data acquisition normalization steps. Also, these results indicate that traditional missing value imputation methods provide no benefit over simple sampling based strategies. Note that common strategies not evaluated included probabilistic PCA. Filling with the mean or median was briefly explored, but did not lead to any performance benefits (Data not shown).

### 2.4. Fold-Change Transformation Showed Consistent Superiority to Other Data Transformation Methods

Omics (e.g., transcriptomics, metabolomics, fluxomics) data often follows a log-normal distribution whereby most components are clumped towards the lower range of values while a few components are found with high abundances that stretch towards the tail. Many statistical methods assume normality (e.g., Students’ T-test) or perform poorly with non-normalized/non-standardized data (e.g., PCA). In particular, deep learning has been shown to perform poorly when the range of magnitude of the data exceeds 0 to 1 or −1 to 1. The following data normalization methods were explored:No normalization (None),Projection to the range 0 to 1 (Proj),Log normalization followed by projection (LogTransProj),Standardization followed by projection (StandProj),Log normalization followed by standardization followed by projection (LogTransStandProj),Log fold change using bases of 2, 10, or 100 for the base term (FCLogX where X indicates the base used).

The normalization methods were applied either offline to the entire dataset (Off) or online on a sample-by-sample basis (On) prior to training and evaluation (See methods).

The influence of transformation methods were tested. In most cases, It was observed that the transformation method had the largest impact on classification accuracy and training speed (Figure 3C,D and Appendix A). In general, fold change transformations (i.e., FCLog10 or FCLog100) showed the highest classification accuracy on the training and test splits followed by OffLogTransProj, OffProj, OffStandProj, OnLogTransProj, OnProj, OnStandProj, and None (Figure 3C,D and Appendix A). In most cases, fold change transformations showed an order of magnitude faster convergence speed than all other transformation methods. The inferior performance of standardization alone and the superior performance of logarithmic transformation alone or in combination with standardization indicates that standardization does not improve classification accuracy nor training speed. The inferior performance of On normalization compared to Off normalization methods indicates that normalization on a dataset basis as opposed to a sample basis is required for optimal performance in deep learning classification using Omics data as is standard practice in other machine learning algorithms.

In summary, the combination of biomass normalization (i.e., BN), data calibration (i.e., Ratios or Concs), sampling based missing value imputation (i.e., Sampling or MARs), and fold change transformation (i.e., FCLog10 or FCLog100) demonstrated superior classification, generalization, and convergence speed to all other methods tested on datasets 1 and 2. Based on the results reported for the classification experiments, the influence of a subset of data pre-processing methods, which included Concs and ConcsBN biomass normalization and data calibration methods, Sampling missing value imputation method, both well-performing and control (i.e., None, OnProj, OffProj, FCLog100, and FCLog10) transformation methods, and reconstruction loss function were used to evaluate reconstruction and joint reconstructions and labeling task performance in a Bayesian or non-bayesian setting with or without supervision.

### 2.5. Reconstruction Accuracy Was Highly Dependent upon the Evaluation Metric Used

The most accurate metric to determine the similarity between Omics datasets that are biologically meaningful is an open area of research [23,24]. Features in Omics data are highly correlated, indicating that only a subset of features provide non-redundant information and hence have a greater influence in determining the phenotype of the organism. In metabolomics, informative features (i.e., metabolites) often include signaling molecules that are involved in enzymatic and transcription regulation, energy or nitrogen carriers whose ratios drive many metabolic reactions via thermodynamics, and cofactors and structural scaffolds that are required for proper enzymatic activity. Informative metabolites also span several orders of magnitude in concentration, which necessitates the need to properly weight lower abundant metabolites relative to higher abundant metabolites. There is a plethora of similarity metrics described in the literature each with their strengths and weaknesses in addressing the challenges of computing a similarity score between metabolomics datasets. Therefore, the influence of loss functions for training deep-learning models to reconstruction metabolomics data was tested. A well-balanced set of metrics including divergence/correlation, distance, and magnitude scores (see methods) were included when evaluating dataset similarity to facilitate a discussion of the influence of loss function on reconstruction accuracy (i.e., how similar the network input and output are) and training convergence speed. An overview of the reconstruction task and evaluation procedure are presented in Figure 4A,B.

The influence of data preprocessing and loss function on reconstruction accuracy was dependent upon the metric used (Figure 4C,D, Appendix A), but showed strong consistency within each class of metrics. For divergence/correlation metrics, OnProj with MSE, MAPE, or MAE loss functions had the highest (i.e., best) cosine similarity and Pearson’s R score, followed by FCLog100 and FCLog10 with MSE or MAE loss functions, respectively. Interestingly, FCLog100 and FCLog10 with MAPE loss function and None with all loss function were the worst as evaluated by cosine similarity and Pearson’s correlation coefficient, respectively. For distance metrics, OffProj with all loss functions tested had the lowest (i.e., best) Euclidean, Manhattan, Jeffreys and Matusita, and Logarithmic distances, followed by OnProj, FCLog100, and FCLog10 with all loss functions tested, respectively. None with all loss functions tested had the lowest Euclidean, Manhattan, Jeffreys and Matusita, and Logarithmic distances as expected. For magnitude metrics, FCLog100 and FCLog10 with all loss functions tested had the lowest (i.e., best) percent difference, followed by OnProj and None with all loss functions tested, and then OffProj with all loss functions tested, respectively. In particular, FCLog100 and FCLog10 with MAPE loss function appeared superior to MSE or MAE loss functions as evaluated by the percent difference. Minimal differences in training speed were observed across data-preprocessing steps and loss functions.

Based on the classification results found earlier and the discrepancy between reconstruction accuracy metrics, it was suspected that the divergence/correlation metrics tested here do not meaningfully correlate with reconstruction accuracy. Instead, greater weight should be given to the distance and magnitude class of metrics tested here. In particular, if overall distance is the primary reconstruction metric one seeks to maximize, a combination of OffProj with MSE, MAE, or MAPE loss is recommended based on the results described above. If magnitude of each feature is the primary reconstruction metric one seeks to maximize, a combination of fold-change transformation with MAPE loss is recommended based on the results described above.

However, several general trends were observed regardless of the reconstruction metric used. First, there appeared to be a slight advantage to using base 100 compared to base 10 for the fold-change transformation. This appears to indicate that including an extra magnitude of change in the range of the dataset prior to clipping the maximum and minimum can lead to better reconstruction accuracy. Please note that only a small fraction of metabolites levels would be clipped when using a base 100-fold change whereas a much larger portion would be clipped when using a base 10-fold change. Second, biomass normalization did not appear to make any significant difference to the reconstruction quality as judged by the different metrics for reconstruction accuracy nor time to convergence. This appears to indicate that, similar to classification accuracy, reconstruction accuracy is not influenced by metabolite level differences derived from sample biomass discrepancies when the data are calibrated.

### 2.6. (Semi-/Un-)Supervised Latent Space Disentanglement of Class and Style Attributes

Semi-supervised or unsupervised clustering and classification of data are a cornerstone analysis in the life sciences for both quality control and biological discovery. Classical methods include PCA and PLS which use a single linear transformation to project high-dimensional data into a latent space. Deep generative modeling can be viewed as a generalization of classical methods where a single linear transformation is replaced by a non-linear deep neural network. Recent efforts have demonstrated the utility of deep generative models to transform data into a latent space that either directly disentangles data labels (i.e., classes) from other data attributes (i.e., style) [25] or to facilitate subsequent clustering algorithms that use the latent space directly [26,27]. To better understand the influence of data preprocessing method and reconstruction loss for the application of deep generative models to directly disentangle data labels from other data attributes in the latent space, a series of supervised, semi-supervised, and unsupervised joint classification and reconstruction experiments were run. An overview of the joint reconstruction and classification task and related subtasks are given in Figure 5A–C.

In the supervised setting, the addition of a joint classification objective was not found to decrease the model’s reconstruction performance nor impact the influence of different combinations of data preprocessing steps and reconstruction loss functions from those found previously (Figure 5D,E, Appendix A), but classification accuracy was noticeably lower than when a dedicated model (with far fewer parameters) was optimized only for the classification tasks (see previous sections on classification accuracy). Furthermore, it was discovered that the magnitude of the reconstruction and classification losses needed to be tuned to achieve optimal reconstruction and classification accuracy (Figure 5D,E, Appendix A). When optimal hyperparameters were used, similar to previous findings for the classification tasks, fold-change transformations (i.e., FCLog10 and FCLog100) showed superior classification accuracy compared to OnProj, OffProj, or None for any combination of reconstruction loss function. No major differences in training convergence time were observed when overall reconstruction and classification accuracies were taken into account.

In the semi-supervised setting, the percentage of supervision had a large effect on the classification results and required training time to convergence. High classification accuracies (i.e., greater than 90%) could be observed at a supervision percent at or above 25% (i.e., 100, 50, and 25%), but deteriorated rapidly when moving to 10, 5, and 1% supervision (Figure 5F,G and Appendix A). To reach classification convergence, a doubling of the training iterations was required (i.e., from 1 × 10^5^ to 2 × 10^5^ iterations). In general, fold-change transformations (i.e., FCLog10 and FCLog100) in combination with MAPE reconstruction loss showed superior classification accuracy compared to MSE or MAE reconstruction loss function at any level of supervision. The weight of the KL divergence terms was also found to influence the classification accuracy at all levels of supervision, where increasing the weight from 1 to 30 lead to a rapid drop in classification performance (data not shown).

In the Unsupervised setting, the influence of the number of discrete and continuous nodes on reconstruction accuracy and disentanglement of data class and style attributes was first assessed. In general, reconstruction accuracy appeared insensitive to the exact distribution of node types when normalized to the total amount of information stored in the latent space (Figure 5H,I and Appendix A). The effect of the preprocessing method and loss function was consistent with what was found for reconstruction accuracy previously (Figure 5H,I and Appendix A). The disentanglement of class label and style information applied to the continuous and discrete nodes was assessed by traversing the latent space and comparing the reconstruction distance to the ground truth labels (see Methods Section 3.3). In general, class labeling information did not appear to be preferentially applied to the discrete nodes, but instead appeared to be intermixed with continuous nodes regardless of the number or distribution of discrete and continuous nodes (Figure 5C, Appendix A). FCLog10 and FCLog100 was found to have qualitatively improved ability to separate and allocate class labels along the discrete nodes compared to OffProj for any combination of loss function used (Figure 5C, Appendix A). The intermixing of class label information between both continuous and discrete nodes was consistent with the semi-supervised experiments described above.

In the Unsupervised setting, the variability of reconstruction and disentanglement of data class and style attributes performances was next assessed. Networks with a class label matched number of discrete nodes and a single continuous nodes were trained (*n* = 12 replicates, Figure 5F,G and Appendix A, see Methods for details). A low variability in the reconstruction accuracy (RSD < 0.02, *n* = 12) as measured by Euclidean distance for networks trained with either MSE and MAPE loss functions and as measured by percent difference for networks trained with MAPE loss function was found (Appendix A). However, a high variability in the reconstruction accuracy (RSD >> 0.02, *n* = 12) as measured by percent difference for networks trained with the MSE loss function were found (Appendix A). This indicates that loss function can influence the variability of the trained network depending upon the evaluation metric used. The accurate assignment of class label information to the discrete nodes was assessed by (see Methods Section 3.3). A high variability in how class label information was distributed among the discrete nodes was found as indicated by the number of unique labels assigned to the discrete nodes (Table 1, Appendix A). The high variability in class label information distribution among both discrete and continuous nodes indicates that for high-dimensional Omics data, the ability of the network to learn to disentangle class and style attributes and allocate those attributes to discrete and continuous nodes is extremely challenging without the use of labels to guide the training process.

## 3. Methods

### 3.1. Datasets

Dataset 1 (referred to as “IndustrialStrains” in the figures, tables, and text) was derived from [28]. Dataset 2 (referred to as “ALEsKOs” in the figures, tables, and text) was derived from [29]. Absolute metabolite concentrations (referred to as “Concs” in the figures, tables, and text) from both datasets are as reported in their respective articles; ion intensities and peak height ratios (referred to as “Intensities” and “Ratios” in the figures, tables, and text, respectively) were not reported in the articles and are provided here for reproducibility. The samples ran using the “long gradient” from Dataset 1 were used for training and the samples ran using the “short gradient” from Dataset 1 were used for testing. Dataset 2 was class balanced by retaining the unevolved knockout strains and using only two of the endpoint replicates for training; a third endpoint was used for testing.

### 3.2. Data Preprocessing

Normalization was performed by dividing the measured Concs, Intensities, or Ratios by the measured biomass for the sample at the time of sample collection.

Missing values were imputed either by sampling, mass action ratio (MARs) sampling, using the AMELIAII algorithm [30] using the same parameters as described in each datasets original publications, substituting zero for the missing value, or using the mean of all biological and technical replicates for the missing value. Sampling was performed by randomly selecting a single replicate for each metabolite based on a uniform distribution. A genome-scale metabolic reconstruction of metabolism [31] was used to derive MARs for each reaction in the model. A MAR is calculated using the following formula:$$\mathrm{MAR}=\frac{R_{1}^{r_{1}}R_{2}^{r_{2}}}{P_{1}^{p_{1}}P_{2}^{p_{2}}}$$
where *R* is the reactant, *P* is the product, and *r* and *p* are the stoichiometric coefficients of the *r* and *p*, respectively. MARs were calculated by randomly selecting a single replicate for each metabolite based on a uniform distribution. Metabolites that are involved in the reaction, but were not measured in the dataset were omitted except for Values for Phosphate, water, dihydrogen, oxygen, carbon dioxide, and hydrogen were estimated as 1.0, 55.0 × 10^−3^, 34.0, 55.0, 1.4, and 1.0, respectively. All other metabolites that were involved in a reaction but were not measured were estimated as 1.0. MARs were constrained to the range of 1 × 10^3^ and 1 × 10^−3^. MARs with less than 50% measured metabolite coverage were omitted.

Cons, intensities, and ratios were normalized offline on a dataset basis (Off) where all parameters used for normalization in the training set were applied to the test set or online on a sample per sample basis (On). Normalization methods tested included projection of the concs, intensities, and ratios to the range 0 and 1 (Proj) using either the minimum and maximum of the sample (OnProj) or dataset (OffProj), log 2 transformation followed by projection (LogTransProj), standardized to the mean and standard deviation (Stand), fold-change transformed relative to a control using log bases of 10, 20, or 100 and value clipping over the interval of −1 and 1 to constrain values to the interval −1 and 1 (FCLogX where X is the base used), and various combinations of the above as described in the text and figures.

### 3.3. Model and Training Details

Two model architectures were used in this study: a classifier network and a reconstruction network. The classifier network was a single hidden layer fully connected feedforward neural network where the input layer was the size of the number of inputs (i.e., metabolites or MARs), and hidden layer was eight nodes, and the output layer was the size of the number of classification labels (i.e., strains). The ADAM solver [2] with an alpha of 1 × 10^−4^ was used for training. The classifier network was trained for 1 × 10^5^ iterations for both datasets using a cross-entropy (XEntropy) loss function. Multi-class micro accuracy (AccuracyMCMicro) and multi-class micro precision (PrecisionMCMicro) were calculated to determine model performance. Gradient clipping of 10, batch size of 64, leaky ReLU activation function [32] with a slope of 0.01 for values less than 0, and He weight initialization [33] was used for all datasets and models.

The reconstruction network included a single hidden layer fully connected feedforward encoder and decoder neural networks each with a hidden layer size of 16. The encoder network had an input layer that was the size of the number of inputs (i.e., metabolites or MARs), and an output layer (i.e., latent layer) whose size depended upon the experiment (detailed below). The decoder network had an input layer (i.e., latent layer) whose size depended upon the experiment (detailed below), and an output layer that matched the size of the encoder input layer.

The reconstruction network was operated as a variational autoencoder (VAE) [34] consisting of a continuous latent space or joint variational autoencoder (JVAE) [25] consisting of both continuous and discrete latent spaces. The output of the VAE encoder was split into an equal number of mean and variance encodings which were used to calculate the latent encoding via the re-parameterization trick for a Gaussian distribution. The input of the decoder matched the size of the encoder latent space. The output of the JVAE was split into mean, variance, and logit encodings. The mean and variance encodings were used to calculate the continuous latent space via the re-parameterization trick for a Gaussian distribution, while the logit encodings were used to calculate the discreet latent space using the re-parameterization trick for the Concrete distribution. The input of the decoder matched the size of the encoder latent space. The exact sizes of the encodings and latent spaces are provided in the figures and Appendix A for each of the different experiments. The ADAM solver [2] with an alpha of 1 × 10^−5^ was used for training. Unless otherwise mentioned, the reconstruction networks were trained for 1 × 10^5^ iterations with an incremental increase of beta from 0 to 1 over 2.5 × 10^4^ iterations for both datasets using either a mean squared error (MSE), mean absolute error (MAE), or mean absolute percent error (MAPE) loss for reconstruction accuracy, and Kullback–Leibler divergence for latent distributions. Capacitance was not explored in this study. During supervised experiments, an additional classification loss using cross-entropy was applied to the discrete latent space where the size of the latent space equaled the number of classification labels. During semi-supervised experiments, the cross-entropy loss was applied randomly with a frequency based on the percent of supervision (e.g., 50% supervision implies that the cross-entropy was applied on average 50% of the time). During unsupervised experiments, unless otherwise noted, encoding layers with distributions of four continuous and four discrete nodes, two continuous and ten discrete nodes, or two continuous and n discrete nodes (where n was equal to the number of class labels for the dataset) were used. Cosine similarity (CosineSimilarity), Pearson’s R value (PearsonsR), Euclidean distance (EuclideanDist), Manhattan distance (ManhattanDist), Jeffreys and Matusita distance (JeffreysAndMatusitaDist), Logarithmic distance (LogarithmicDist), and Percent difference (PercentDifference) were calculated to determine model reconstruction performance. During supervised experiments, multi-class micro accuracy (AccuracyMCMicro) and multi-class micro precision (PrecisionMCMicro) were also calculated to determine classification performance. Latent traversals for continuous nodes were performed by sample 16 even steps from 0.05 to 0.95 from an inverse normal distribution for a single node while keeping all other nodes constant (i.e., 0). Latent traversals for discrete nodes were performed by fixing one node to a value of 1 while keeping all other nodes constant (i.e., 0). Gradient clipping of 10, batch size of 64, leaky ReLU activation function [32] with a slope of 0.01 for values less than 0, and He weight initialization [33] was used for all datasets and models.

All models were trained using an Intel Xeon Gold 6230 processor with 128 GB of RAM and an NVIDIA Titan RTX GPU with 28 GB of memory.

### 3.4. Hyperparameter Tuning

Model and training hyperparameters that were tuned prior to model training and testing included classification and reconstruction hidden layer sizes and depth, training iterations, batch size, and learning rate. The hidden layer sizes and depth were chosen to minimize overfitting and to constrain the top performing model to approximately 95% maximum accuracy for classification and approximately 0.95 maximum Pearson’s R for reconstruction similarity. Several batch sizes were explored including 16, 32, 64, 128, and 256 with 64 providing the best tradeoff between performance and training speed. Learning rates explored included 1 × 10^−2^, 1 × 10^−3^, 1 × 10^−4^, and 1 × 10^−5^ and were chosen based on their training stability and final classification and/or reconstruction accuracy.

### 3.5. Code Availability

All models and algorithms were written in C++17 using the General and Tensor Eigen libraries and CUDA APIs. The code is available at https://github.com/dmccloskey/evonet (access on 14 February 2022). All analyses and plotting routines were written in python 3.6 using the numpy and matplotlib libraries. The code is available at https://github.com/dmccloskey/EvoNetPyScripts (access on 14 February 2022).

## 4. Conclusions

Data pre-processing has a major role in any data analysis workflow, and in particular, high dimension and heterogeneous data such as that found in modern high throughput life science experiments. A representative number of methods that span the entire data pre-processing pipeline including those for data calibration, biomass normalization, missing value imputation, and data normalization were assessed (Figure 1A). Two representative datasets of real world settings that included batch effects and uneven distribution of replicates were utilized (Figure 1B,C). The influences of data pre-processing and dataset were first assessed for classification tasks and then the influences of data-processing, dataset, loss function, and evaluation metrics were assessed for reconstruction and class label and style disentanglement tasks (Figure 3, Figure 4 and Figure 5). The results from these experiments can be summarized in Box 1.


Box 1Summary of data-preprocessing evaluations.
Differences in biomass can be accounted for using deep learning, but data calibration is needed to account for sample handling and data acquisition effects.A sampling strategy to account for missing values provided superior performance to other missing value imputation methods.Log fold-change transformation with an optimized base provided superior performance to alternative data normalization methods.The optimal combination of data normalization and loss function is dependent upon the reconstruction evaluation metric chosen: Offline project of the dataset to between 0 and 1 to improve numerical
stability in combination with MSE or MAE loss was superior for distance-based metrics, while log fold-change transformations in combination with MAPE loss was superior for feature magnitude metrics.Log fold-change transformations in combination with MAPE loss provided superior performance for joint reconstruction and classification tasks.Unsupervised disentanglement of Omics latent features is a challenging task that requires some level of semi-supervision or prior information to work well.



This study focused on the use of metabolomics data as a representative life science data type. Future work could look to extend the results found here to other life science data types (e.g., RNA sequencing). This study also focused on deep learning which is much less studied in the analysis of Omics data compared to more traditional machine-learning methods. The experiments described here highlight the unique data-preprocessing requirements for classifying and reconstructing Omics data when using deep learning. Future efforts will also look to incorporate this work into ongoing efforts in the development of deep-learning frameworks that take multiple Omics modalities as inputs.

## Figures and Tables

**Figure 1 metabolites-12-00202-f001:**
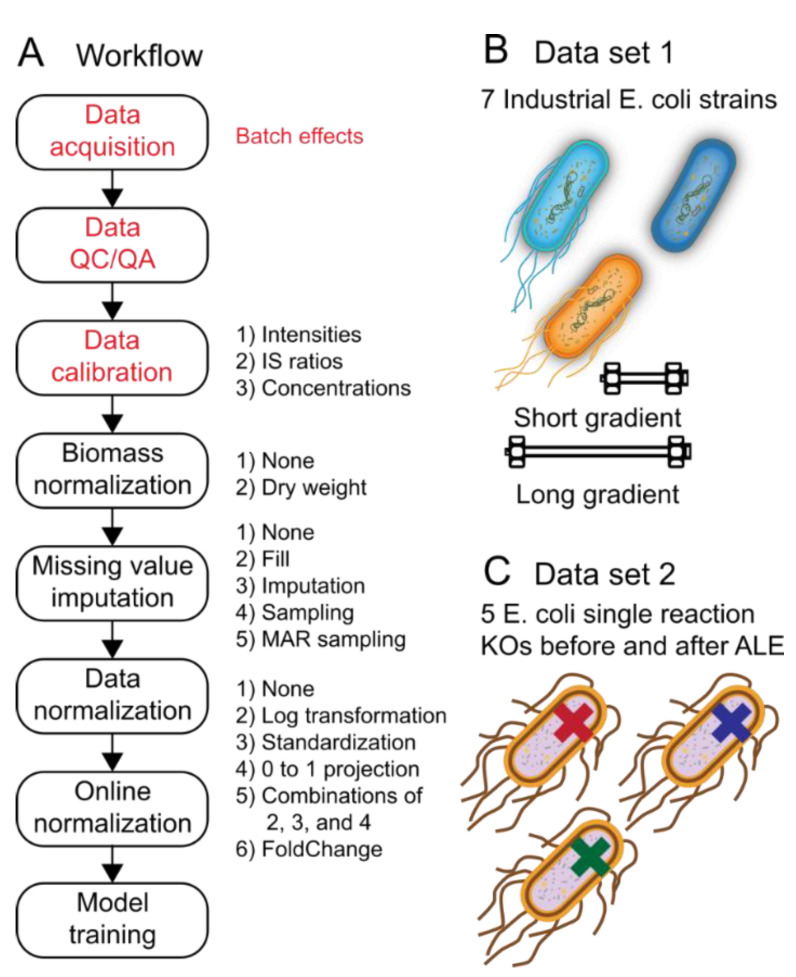
Data preprocessing workflow for metabolomics. (**A**) Workflow overview. Red lettering in the workflow diagram indicates steps that lead to so-called “Batch effects” between metabolomics runs. Black lettering to the right of each workflow box indicates preprocessing options that were explored in this study for the corresponding data preprocessing workflow steps. (**B**,**C**). Datasets 1 and 2 used for the classification task (see Methods Section 3.1). (**B**). The dataset included 7 different strains of *E. coli* where the task was to predict the correct strain from metabolite levels. Importantly, the data were run twice (once on two different instruments) resulting in batch effects between the two runs. (**C**). The dataset included 5 different single-knockout (KO) strains of *E. coli* where the task was to predict the correct knockout from metabolite levels.

**Figure 2 metabolites-12-00202-f002:**
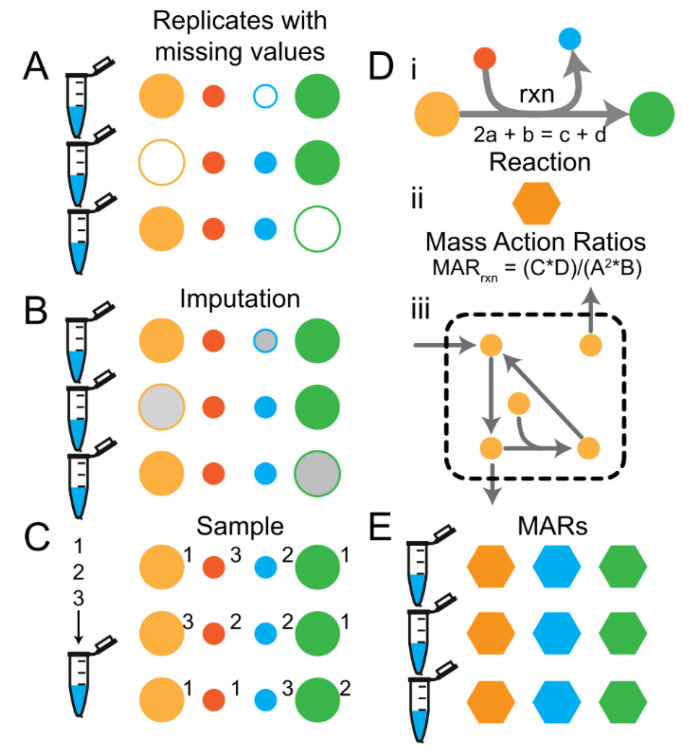
Missing value imputation methods. (**A**) Schematic of 3 replicate samples (eppi tubes) with 4 metabolite data points (circles). Filled circles indicate measured values and open circles indicate missing values. (**B**) Missing values are usually imputed (grey filled circles) prior to downstream machine-learning tasks. (**C**) Alternatively, the values for each metabolite from all replicates can be pooled and sampled to generate values for replicates on the fly. The numbers next to each circle indicate the replicate number from which the value came. (**D**) Overview of the procedure for calculating Mass Action Rations (MARs) for metabolomics data given a metabolic network stoichiometry. Please note that the number of features per sample changes from the number of metabolites measured to the number of metabolic reactions in the metabolic network. (**E**) MARs can be calculated by sampling the metabolite values from each replicate.

**Figure 3 metabolites-12-00202-f003:**
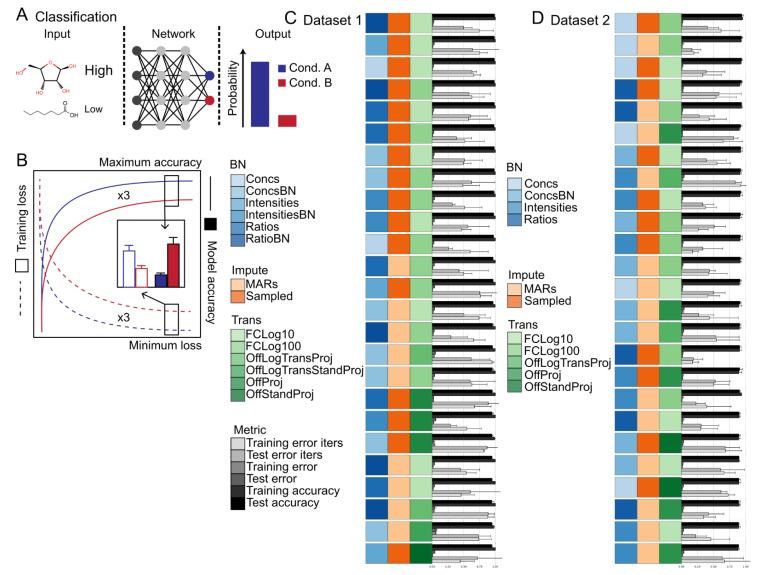
Summary of the classification task and evaluation procedure. (**A**) Schematic of the machine-learning classification task using metabolite levels as input and sample labels as output. (**B**) Diagram of the evaluation procedure. Training and test losses are depicted in blue and red. Summary of classification results for Dataset 1 (**C**) and Dataset 2 (**D**). The bars represent the model training iterations to minimum loss function scores, the model minimum loss function scores, and the model maximum accuracy (*n* = 3 training runs). Error bars represent the standard deviation (*n* = 3 training runs). Model training iterations to minimum loss and model loss scores are scaled from 0 to 1. The data are sorted in descending order based on model accuracy. The top 25 models are shown. Source data are provided in Appendix A Abbreviations: BN—Biomass normalization; Impute—Imputation method; Trans—Transformation method.

**Figure 4 metabolites-12-00202-f004:**
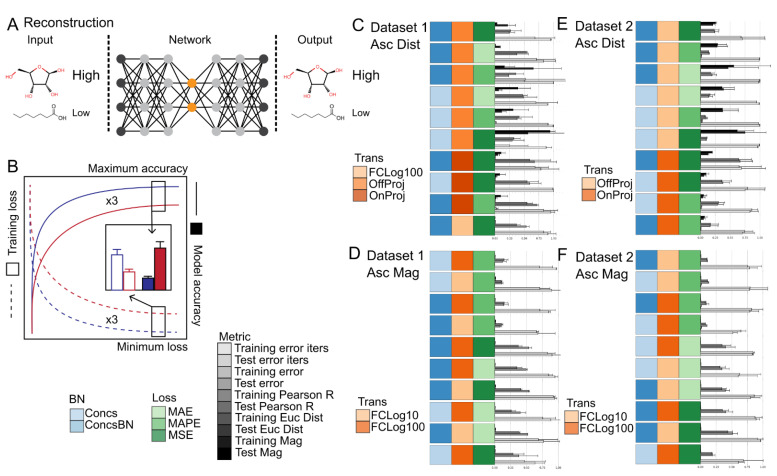
Summary of the reconstruction task and evaluation procedure. (**A**) Schematic of the machine-learning reconstruction and joint reconstruction and classification tasks using metabolite levels as input and sample labels as output. (**B**) Diagram of the evaluation procedure. Summary of Reconstruction results for Dataset 1 (**C**,**D**) and Dataset 2 (**E**,**F**). The bars represent the model training iterations to minimum loss function scores, the model minimum loss function scores, and the model metric scores (*n* = 2–3 training runs). Error bars represent the standard deviation (*n* = 2–3 training runs). Reconstruction metrics for Pearson’s R, Euclidean distance, and Absolute percent difference are shown. Model training iterations to minimum loss, model loss scores, Euclidean distance, and Absolute percent difference are scaled from 0 to 1. The Imputation method of Sampling was used for all models. The data are sorted in ascending order based on Euclidean distance (**C**,**E**) and Absolute percent difference (**D**,**F**). The top 10 models are shown. Source data are provided in Appendix A. Abbreviations: BN—Biomass normalization; Impute—Imputation method; Trans—Transformation method; Loss—Loss function; MSE—Mean squared error; MAE—Mean absolute error; MAPE—Mean absolute percent error.

**Figure 5 metabolites-12-00202-f005:**
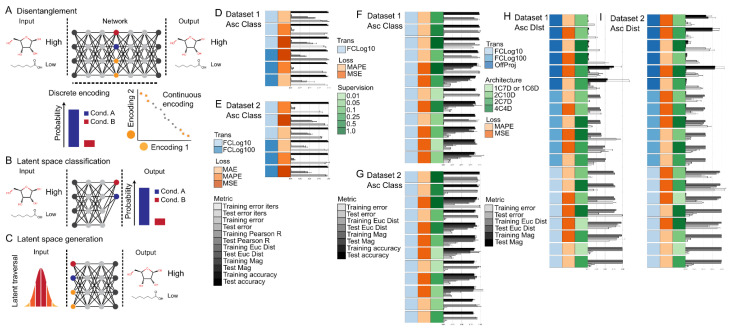
Summary of the disentanglement tasks. (**A**) Schematic of the machine-learning joint reconstruction task using metabolite levels as input and sample labels and metabolite levels as output. The reconstruction task entailed encoding the input features into a lower dimensional latent space and then decoding the compressed representation back to the original input features. A more meaningful latent space is one where factors of variation in the input features are disentangled into specific regions of the latent space. (**B**) Schematic of the direct latent space classification subtask. The classification task entailed capturing the input labels directly in the discreet encodings of the latent space. (**C**) Schematic of the latent traversal and reconstruction similarity subtask. The latent traversal and reconstruction similarity subtasks entailed traversing the 95% confidence intervals of the continuous encodings and one hot vectors of the discreet encodings, then decoding and evaluating the resulting reconstruction against randomly sampled inputs with known labels. Summary of the joint reconstruction and classification task results for Dataset 1 (**D**) and Dataset 2 (**E**). Summary of the influence of supervision on the direct latent space classification subtask for Dataset 1 (**F**) and Dataset 2 (**G**). Summary of the influence of latent space architecture on reconstruction accuracy for Dataset 1 (**H**) and Dataset 2 (**I**). The bars represent the model training iterations to minimum loss function scores, the model minimum loss function scores, and/or the model metric scores as specified per the legend next to each figure panel. Error bars represent the standard deviation. *n* = 2 for all models shown except for models with an architecture of 1C6D and 1C7D (*n* = 12) shown in (**H**,**I**). Model training iterations to minimum loss, model loss scores, Euclidean distance, and Absolute percent difference are scaled from 0 to 1. Top 6 models sorted by classification accuracy are shown in (**D**,**E**); Top 10 models sorted by classification accuracy are shown in (**E**,**F**); All models sorted by reconstruction distance are shown in (**G**,**H**). Biomass normalization of ConcsBN and Imputation method of Sampling was used for all models. Source data are provided in Appendix A. Abbreviations: Trans—Transformation method; Loss—Loss function; Supervision—Percent supervision used for classification; Architecture—Model architecture used where xC is the number of continuous valued nodes and xD is the number of discrete valued nodes in the latent space; MSE—Mean squared error; MAE—Mean absolute error; MAPE—Mean absolute percent error.

**Table 1 metabolites-12-00202-t001:** The number of unique class labels applied to each of the discrete node layers (n = 7 for Dataset 1 and n = 6 for Dataset 2) where Dataset 1 has 7 unique classes and Dataset 2 has 6 unique classes. ConcsBN, Sampling, and FCLog10 preprocessing was used for all networks. The latent space for Dataset 1 consisted of 7 discrete nodes and 1 continuous node while the latent space for Dataset 2 consisted of 6 discrete nodes and 1 continuous node. The average and standard deviation are based on n = 12 replicate networks.

Dataset	Split	Type	AVE (Euc)	STDDEV (Euc)	AVE (Perc Diff)	STDDEV (Perc Diff)
Dataset 1 (Industrial)	Train	MAPE	4.917	0.493	4.750	0.722
		MSE	4.250	1.090	4.500	0.500
	Test	MAPE	4.417	0.954	4.500	0.866
		MSE	4.417	0.954	4.833	0.687
Dataset 2 (ALEsKOs)	Train	MAPE	3.000	0.000	3.750	0.595
		MSE	3.750	1.010	4.000	0.707
	Test	MAPE	3.250	0.433	4.167	0.986
		MSE	4.167	0.553	3.833	0.799

## Data Availability

The data presented in this study are available in the article and Appendix A.

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
