# Peer review of "A Comprehensive Evaluation of Metabolomics Data Preprocessing Methods for Deep Learning"

_metabolites, 2022, doi:10.3390/metabo12030202_

Round 1

Reviewer 1 Report

This is an interesting work in which the efforts of various strategies used in the data preprocess steps of metabolomics on the performance of downstream deep learning were tested and evaluated. I think this work would be helpful for not only the data processing in metabolomic studies, the methodology can be extended to other omics of which the data is also in high dimension and non-normally distributed. I recommend acceptance of this manuscript after the typographical errors are corrected and the missing Figure 3 is provided.

Author Response

We thank the reviewers for their positive comments. We have re-edited the manuscript to correct typographical errors. We have also restored figure 3.

Reviewer 2 Report

This is an interesting article that investigates the impact of metabolomic data preprocessing algorithms on the performance of deep learning models.  But the article and its code repository are poorly prepared. Therefore, the recommendation is major revision to improve the article and its code repository.
(1) The authors need to define the reconstruction task and  the disentanglement task clearly.
(2) Where is Box 1?
(3) Figure 3 is missing.
(4) There are numerous figure citation errors in this manuscirpt, which  seriously affects the reading experience of the article.
(5) The presentation of the results of the reconstruction and disentanglement tasks needs to be improved to make it easier for the reader to understand.
(6) How the training, validation and testing sets for the neural networks were constructed from the IndustrialStrains and ALEsKOs datasets was not clearly explained.
(7) The precise network architectures  corresponding to the three tasks need to be given.
(8) The source code library corresponding to the article needs to be improved, including Readme files, examples, test data, models, documentation, etc. This allows users to quickly get started with the methods developed by the authors.

Author Response

We thank the reviewers for their supportive critique of the manuscript. We have addressed each of the reviewer's comments as detailed below.

(1) The authors need to define the reconstruction task and the disentanglement task clearly.

We thank the reviewer for highlighting this inadequacy. We have added an expanded description of the different tasks in the caption for figure 5 as well as in section 2.5.

(2) Where is Box 1?

Box 1 was somehow omitted from the initial version. It has now been restored.

(3) Figure 3 is missing.

Figure 3 was somehow omitted during the upload of the manuscript. It has now been restored. 

(4) There are numerous figure citation errors in this manuscript, which seriously affects the reading experience of the article.

We have gone through the manuscript and corrected all figure citations.

(5) The presentation of the results of the reconstruction and disentanglement tasks needs to be improved to make it easier for the reader to understand.

We thank the reviewer for this constructive criticism.  We have revised the caption for Figure 5 with an improved description of the results presented in panels D to I.

(6) How the training, validation, and testing sets for the neural networks were constructed from the IndustrialStrains and ALEsKOs datasets were not clearly explained.

We have added several sentences to the Methods section 3.1 to provide better clarity on how the training and test splits were constructed. 

(7) The precise network architectures corresponding to the three tasks need to be given.

We have added several sentences to the Methods section 3.3 to clarify the network architectures used and how the overall network template was modified for each of the different tasks.

(8) The source code library corresponding to the article needs to be improved, including Readme files, examples, test data, models, documentation, etc. This allows users to quickly get started with the methods developed by the authors.

We have provided a much-needed cleanup of the repository following the reviewer's suggestions. This included a complete overhaul of the documentation, license, an updated README.rst file, and deployment on ReadTheDocs https://evonet.readthedocs.io/en/latest/?badge=latest. The examples and unit tests have also been cleaned up and noted in the documentation. Basic instructions for building the project using CMAKE have also been provided in the documentation.

Reviewer 3 Report

Metabolomics is routinely applied as a tool for biomarker discovery, mechanism understanding and phenotype explanation during the last decade. The data pre-processing methods of metabolomics are critical for the following statistical analysis and machine learning model establishment.

The authors utilized two datasets to evaluate the pre-processing pipeline. Meanwhile, the author mentioned that the work could be used for the construction of the deep learning model. This study is on a topic of relevance and is of general interest to the readers of Metabolites. I found the processing of the datasets to be well done. However, the description of some very important points was inadequate. It would be appreciated if the authors specifically address each of the following comments in their response. 

  1. It is unclear how the authors’ study non-batch effects and class balances since dataset 1 has only batch effects and dataset 2 has several class imbalances. This could get the readers confused.
  2. In line 34, the formatting error of “advanced optimizers” should be corrected.
  3. In line 272, why do the author choose Pearson correlation analysis rather than others?
  4. In the results and discussion, Figure 3 is missing.
  5. In the conclusion, elaborate more on how this evaluation of metabolomics data pre-processing methods enhance knowledge in the model establishment of deep learning compared with other algorithms.

Author Response

We thank the reviewers for their positive reviews and helpful comments. We have addressed each of the comments as detailed below.

It is unclear how the authors’ study non-batch effects and class balances since dataset 1 has only batch effects and dataset 2 has several class imbalances. This could get the readers confused.

We have added several clauses in section 2.1 where the data sets are introduced which will hopefully provide greater clarity to the reader.

In line 34, the formatting error of “advanced optimizers” should be corrected.

We thank the reviewer for spotting this error. It has now been corrected. 

In line 272, why do the author choose Pearson correlation analysis rather than others?

The reviewers have brought up a good point. Pearson correlation was chosen instead of Spearman or other correlation measures based on rank due to the importance of absolute values when comparing across metabolomic datasets. 

In the results and discussion, Figure 3 is missing.

Figure 3 was somehow omitted during the upload of the manuscript. It has now been restored. 

In the conclusion, elaborate more on how this evaluation of metabolomics data pre-processing methods enhance knowledge in the model establishment of deep learning compared with other algorithms.

We thank the reviewers for this suggestion. We have added several sentences to the conclusion describing the relevance of the experiments described in this study to developing deep learning models for omics data in contrast to more established methods such as PCA, random forest, etc.

Round 2

Reviewer 2 Report

The authors have revised the manuscript according to previous comments, and I have no further comments.